# Selection, biophysical and structural analysis of synthetic nanobodies that effectively neutralize SARS-CoV-2

Tânia F. Custódio[1], Hrishikesh Das [2], Daniel J. Sheward[3,4], Leo Hanke [3], Samuel Pazicky [1], Joanna Pieprzyk[1], Michèle Sorgenfrei[5], Martin A. Schroer [6], Andrey Yu. Gruzinov[6], Cy M. Jeffries[6], Melissa A. Graewert [6], Dmitri I. Svergun[6], Nikolay Dobrev[7], Kim Remans[7], Markus A. Seeger [5], Gerald M. McInerney [3], Ben Murrell [3✉], B. Martin Hällberg [2,8✉] & Christian Löw [1✉]

The coronavirus SARS-CoV-2 is the cause of the ongoing COVID-19 pandemic. Therapeutic neutralizing antibodies constitute a key short-to-medium term approach to tackle COVID-19. However, traditional antibody production is hampered by long development times and costly production. Here, we report the rapid isolation and characterization of nanobodies from a synthetic library, known as sybodies (Sb), that target the receptor-binding domain (RBD) of the SARS-CoV-2 spike protein. Several binders with low nanomolar affinities and efficient neutralization activity were identified of which Sb23 displayed high affinity and neutralized pseudovirus with an IC$_{50}$ of 0.6 μg/ml. A cryo-EM structure of the spike bound to Sb23 showed that Sb23 binds competitively in the ACE2 binding site. Furthermore, the cryo-EM reconstruction revealed an unusual conformation of the spike where two RBDs are in the 'up' ACE2-binding conformation. The combined approach represents an alternative, fast workflow to select binders with neutralizing activity against newly emerging viruses.

[1] Centre for Structural Systems Biology (CSSB), DESY and European Molecular Biology Laboratory Hamburg, Notkestrasse 85, D-22607 Hamburg, Germany. [2] Centre for Structural Systems Biology (CSSB) and Karolinska Institutet VR-RÅC, Notkestrasse 85, D-22607 Hamburg, Germany. [3] Department of Microbiology, Tumor and Cell Biology, Karolinska Institutet, Stockholm 17177, Sweden. [4] Division of Virology, Institute of Infectious Diseases and Molecular Medicine, Faculty of Health Sciences, University of Cape Town, Cape Town, South Africa. [5] Institute of Medical Microbiology, University of Zurich, Zurich, Switzerland. [6] European Molecular Biology Laboratory (EMBL), Hamburg Outstation c/o Deutsches Elektronen Synchrotron (DESY), Notkestrasse 85, D-22607 Hamburg, Germany. [7] European Molecular Biology Laboratory (EMBL) Heidelberg, Protein Expression and Purification Core Facility, 69117 Heidelberg, Germany. [8] Department of Cell and Molecular Biology, Karolinska Institutet, 17177 Stockholm, Sweden. ✉email: benjamin.murrell@ki.se; martin.hallberg@ki.se; christian.loew@embl-hamburg.de

The emerging COVID-19 pandemic imposes a substantial social and economic burden worldwide. The causative agent of COVID-19 that causes, among other symptoms, severe atypical pneumonia in humans was quickly identified as a novel coronavirus and designated as SARS-CoV-2[1–4]. Similar to its closest homolog SARS-CoV-1, SARS-CoV-2 exploits the ACE2 (angiotensin converting enzyme 2) receptor to enter host cells[5–7]. Host cell entry is orchestrated by the viral spike protein that, upon binding to ACE2, mediates fusion of the viral membrane with the host membrane. The ectodomain of the heavily glycosylated spike protein forms a trimer where each protomer consists of a core S2 subunit and a distal S1 subunit. Each S1 subunit comprises a receptor-binding domain (RBD) that can switch between an exposed "up" conformation and a "down" conformation, where the latter is inaccessible for ACE2 binding[8–10]. A number of antibodies against various viral surface proteins are currently in clinical trials[11]. Exposed on the surface of SARS-CoV-2, the spike protein is the major antigenic determinant of the host immune response and one of the main coronavirus drug targets[12,13]. Indeed, plasma from convalescent SARS-CoV-2 patients was suggested to improve the clinical outcome of patients with severe COVID-19[14–16]. Several groups have isolated human antibodies that showed promising neutralization activity against SARS-CoV-2 in vitro and improved the clinical outcome of tested animals in vivo[17–20]. As an alternative to human antibodies, nanobodies (single-domain antibodies) can also be used as therapeutics with the advantage of their small size, increased stability, and superior simplicity in production[21,22]. Recently, several groups showed that nanobodies can inhibit the binding of the spike protein to ACE2 and neutralize the virus[23–25]. Nanobodies are traditionally isolated from immunized camelids; however, the development of libraries of synthetic nanobodies (sybodies) enables a faster and cheaper selection of binders against therapeutic targets[26].

Here, we report the selection of sybodies directed against the RBD of SARS-CoV-2 and demonstrate that they can compete with ACE2 binding and neutralize SARS-CoV-2 spike pseudovirus. In particular, sybody 23 (Sb23) binds to recombinant RBD as well as to the prefusion spike glycoprotein with high affinity and shows very potent neutralization activity. A small angle X-ray scattering (SAXS) model of a RBD–Sb23 complex indicated that Sb23 binds in the vicinity of the ACE2-binding site on the RBD. Lastly, a cryo-EM structure of Sb23 bound to the spike reveals that Sb23 binds in the ACE2-binding site on the RBD in both its "up" and "down" conformation and thereby effectively blocks ACE2 binding. Interestingly, the cryo-EM reconstruction also reveals that the spike bound to Sb23 displays an unusual conformation where two RBDs are in the "up" ACE2-binding conformation.

## Results

**Sybody selection, ELISA and sequencing.** Sybody selections on biotinylated RBD were carried out with the three sybody libraries (concave, loop and convex) following established procedures[26,27]. The enrichment of binders against RBD was closely followed by qPCR with enrichment factors of 100–9000 in the last selection round, indicating strong specific enrichment (Fig. 1a). In general, sybodies that gave rise to a positive ELISA signal for the RBD, did so also for the prefusion spike protein. ELISA signals for the spike protein were on average higher compared to the RBD, which could be due to avidity effects caused by the trimeric nature of the spike protein or an increased biotin density on the surface of the protein. No binding to the negative control protein MBP was observed for any of the binders, indicating RBD-specificity of the selected sybodies (Supplementary Fig. 1). Next, we sequenced 100

clones which were positive in ELISA against RBD and spike covering the three libraries equally well. We obtained 85 unique sybody sequences (Supplementary Data 1). In total, 33 sybodies were derived from the concave, 32 from the loop, and 20 from the convex library. Based on sequence alignment and phylogenetic tree analysis (Fig. 1b), it becomes clear that selected binders cover a large sequence space.

**Sybody purification and affinity analysis.** 62 unique sybodies were expressed in *E. coli*, purified via metal-affinity chromatography to high purity and characterized by analytical gel filtration (Supplementary Fig. 2). We employed a biolayer interferometry (BLI) assay to kinetically characterize the interaction of sybodies with RBD. Biotinylated RBD was immobilized on a streptavidin sensor and on- and off-rates for all purified sybodies were determined at a single concentration (500 nM) yielding affinities ($K_D$) in a range of 5 nM–10 μM (Supplementary Fig. 3). Most sybodies displayed strong binding signals, but many of them exhibited fast off-rates, which is disadvantageous for neutralization activity.

To obtain more accurate affinity data for RBD binding, we selected sybodies 12, 23, 42, 76, 95, and 100 and determined their binding constants by analyzing the on- and off-rates over a large range of sybody concentrations. Resulting affinities are listed in Fig. 2a–f. Tight binding of the selected binders to RBD was further confirmed by thermal shift assays (Fig. 2g, h). In particular, the interaction of Sb23 with RBD increases the melting temperature of RBD by almost ten degrees, which is considerably more than for other binders.

**Sybodies neutralize SARS-CoV-2 spike pseudotyped viruses.** To test whether the selected sybodies can neutralize SARS-CoV-2, we performed a neutralization assay with lentiviral particles pseudotyped with the SARS-CoV-2 spike protein[28]. Thirty-six sybodies were screened for neutralization, identifying eleven capable of neutralizing SARS-CoV-2 at an $IC_{50} < 20$ μg/ml, including six with an $IC_{50} < 5$ μg/ml (Supplementary Fig. 4a–c). No neutralization of VSV-G pseudotyped viruses was evident for any sybody preparation, and a control sybody targeting the human peptide transporter 2 (PepT2) did not show any neutralizing activity. Sybody 23 (Sb23) represented the most potent sybody identified, with an $IC_{50}$ of 0.6 μg/ml (Fig. 3a). The neutralization efficiency of Sb23 was significantly increased by fusing it to an antibody-derived Fc domain. The bivalent Sb23-Fc constructs, displays ~100-fold improved neutralization potential, compared to its monovalent counterpart, with an $IC_{50}$ of 0.007 μg/ml and an affinity towards the RBD in the pM range (Fig. 3a, b). A similar effect was observed for Sb42-Fc (Supplementary Fig. 4d, e).

**Sb23 competes with ACE2 for RBD binding of the SARS-CoV-2 spike protein.** To obtain a deeper understanding of the neutralizing activity of the identified binders, we performed BLI assays to monitor ACE2 binding to immobilized RBD in the presence or absence of Sb23. ACE2 bound RBD with an affinity of 33.4 nM under the stated experimental conditions, but no ACE2-RBD binding was detected in the presence of 150 nM Sb23 over the tested concentration range (Fig. 3c). This could be attributed to either a direct competition for the receptor-binding motif, a steric hindrance, or conformation induced changes upon Sb23 binding.

To confirm this observation in the context of the trimeric spike protein, we developed a competition assay under equilibrium conditions to screen binding of Sb23 to the spike protein in the presence or absence of ACE2 using microscale thermophoresis.

**a**

| | Enrichment Phage display 1 | Enrichment Phage display 2 | Number of ELISA hits RBD/SPIKE (total analysed) | Number of unique binders (total sequenced) | Number of well behaved binders (total analysed) |
|---|---|---|---|---|---|
| **Concave** | 2.9 | 9297 | 62/89 (94) | 33 (34) | 20 (23) |
| **Loop** | 1.2 | 310 | 61/87 (94) | 32 (33) | 20 (24) |
| **Convex** | 0.7 | 118 | 50/87 (94) | 20 (33) | 13 (15) |

**b**

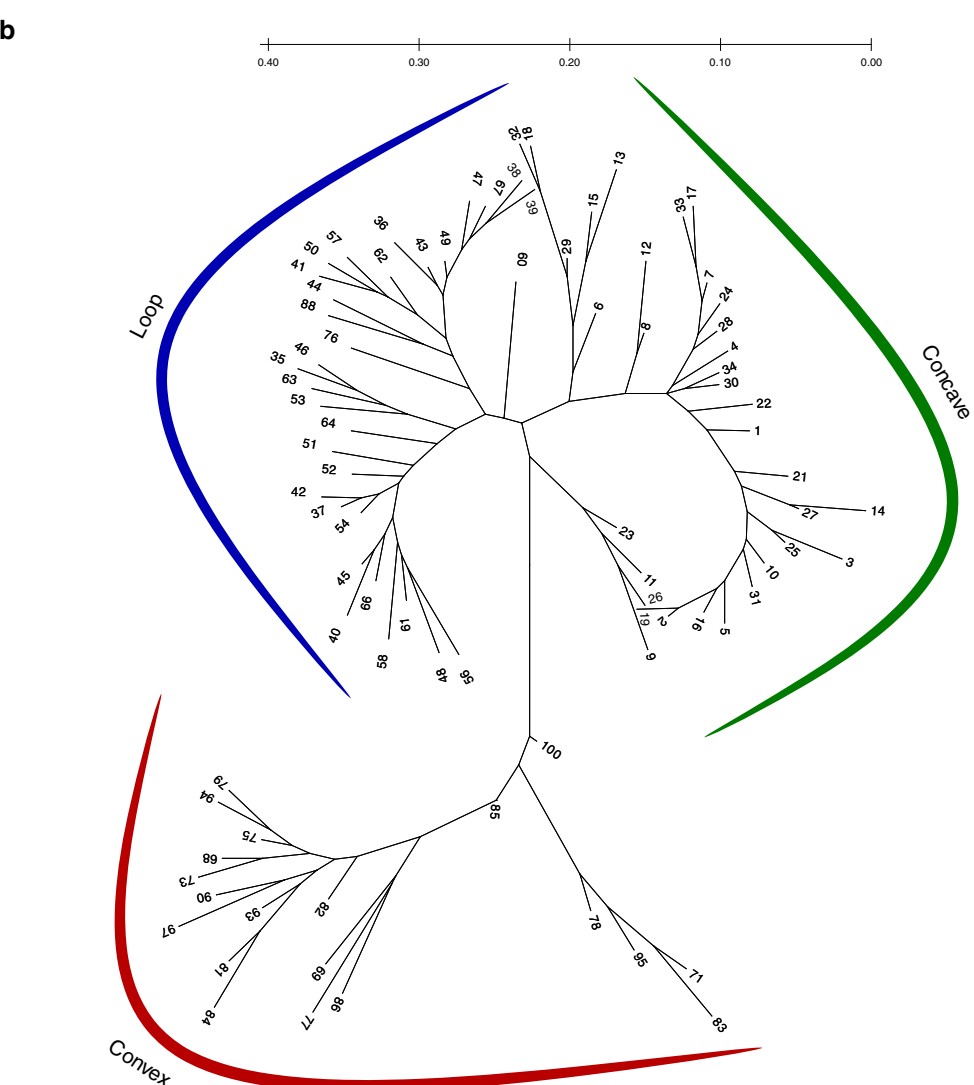

**Fig. 1 Selection of sybodies against SARS-CoV-2 RBD. a** Sybody selection statistics for the three different libraries. **b** Radial phylogenetic tree of all unique binders identified in this study. Sequence alignment was performed using PROMAL3D and the phylogenetic tree was construct using the maximum likelihood (ML) analysis in MEGA. Two sybodies (76 and 88), expected to stem from the convex library, were found to belong to the loop library, presumably due to a spill-over during the selection procedure.

This method is sensitive to changes in size, charge state, and hydration shell, and requires one binding partner to be fluorescently labeled. We monitored binding of the fluorescently labeled Sb23 to the spike protein in the presence of a constant concentration of ACE2. In case the sybody targets a different epitope on the RBD of the spike protein than ACE2, the resulting affinity is expected to be independent of the presence of ACE2. However, our data show that with increasing concentrations of ACE2, the affinity of Sb23 to the spike protein drops 10-fold in the presence of 200 nM ACE2 (Fig. 3d). In conclusion, these data indicate that Sb23 and ACE2 compete for the same or overlapping binding sites on SARS-CoV-2-RBD. At the same time the data also highlight that Sb23 has a significantly higher affinity to the RBD than ACE2 since Sb23 can efficiently replace ACE2 from a preformed complex.

**Characterization of Sb23 and RBD interaction by SAXS.** To shed light on the structural basis of the neutralization activity of Sb23 against SARS-CoV-2, we collected SAXS on the RBD–Sb23 complex (Fig. 4a, b and Table 1). Complex formation is confirmed as judged by the shift of the position of the maximum in p (r) and the increase of the particle size $D_{max}$ upon Sb23 binding. The ab initio low-resolution shape depicts the Sb23 and RBD moieties in the complex (Fig. 4c) and indicates that Sb23 is bound at the extremity of the RBD.

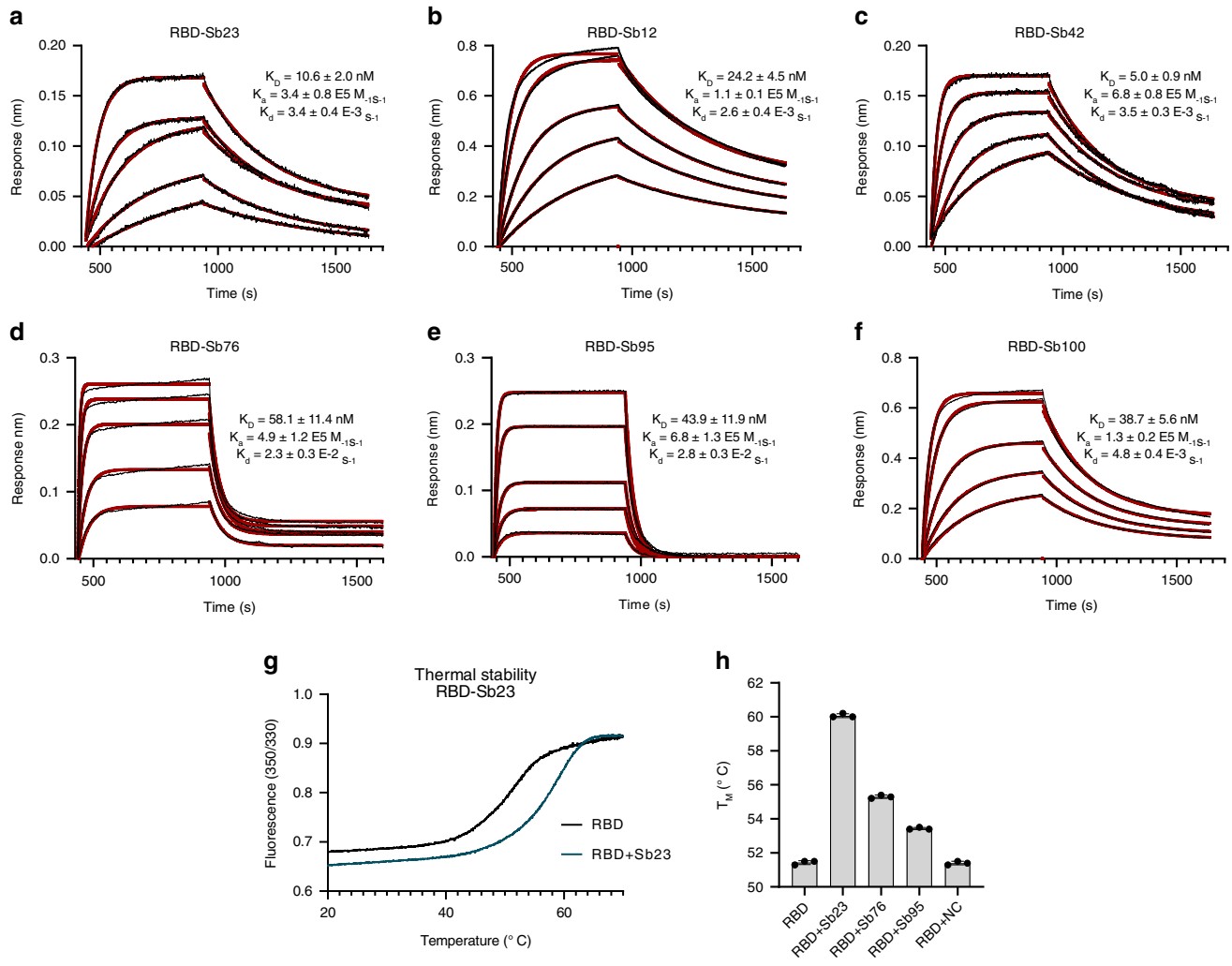

**Fig. 2 Biophysical characterization of identified sybodies.** BLI sensorgrams of immobilized SARS-CoV-2 RBD with 2-fold serial dilution of 75 nM Sb23 (**a**), 188 nM Sb12 (**b**), 50 nM Sb42 (**c**), 250 nM Sb76 (**d**), 250 nM Sb95 (**e**), and 188 nM Sb100 (**f**). Binding curves are colored black and the global fit of the data to a 1:1 binding model is red. Resulting affinities are indicated. **g** Thermal unfolding data of isolated RBD and in complex with Sb23. **h** Resulting melting temperatures of RBD alone and in complex with Sb23, Sb76, Sb95, and a control sybody (NC), selected against the human peptide transporter (hPepT2). Data represent the mean ± SD of three replicate experiments.

To obtain a more detailed understanding, we performed SAXS-based rigid body modeling of the complex between Sb23 and RBD (Fig. 4d). The resulting hybrid rigid body model agrees well with the ab initio shape of the complex (Fig. 4c) and, importantly, in all resulting models, Sb23 is placed next to the ACE2-binding site and binds sidewise to the RBD as expected for a binder from the concave designed library.

**Cryo-EM structure of the SARS-CoV-2 prefusion spike bound to Sb23.** To understand how Sb23 neutralizes SARS-CoV-2, we determined the cryo-EM structure of spike bound to Sb23 (Fig. 5a, b). In the pool of particles that enabled a high-resolution reconstruction, approximately half of the particles had one RBD "up" (1-up) and the other half a conformation with two RBDs "up" (2-up) (Fig. 6a and Supplementary Fig. 5). To the best of our knowledge, the latter conformation has rarely been observed for the SARS-CoV-2 prefusion spike but is commonly observed for the MERS and SARS-1 spikes[29,30].

In both main conformations and for both "up" and "down" spike protomers, Sb23 binds in the inner edge of the ACE2 interaction interface of the RBD thereby effectively hindering

ACE2 binding (Fig. 6b). Modeling the ACE2-spike interaction based on the ACE2-RBD crystal structure (6LZG[31]) of the soluble part of ACE2 bound to SARS-CoV-2 RBD shows that Sb23 hinders binding of ACE2 in both the "1-up" and the "2-up" conformation (Fig. 6b). Interestingly, ACE2 binding to the "up" protomer is hindered in the "1-up" conformation also from Sb23 binding the neighboring "down" protomer (Fig. 6b). This interprotomer-mediated blockage is true also for the corresponding "up" protomer in the "2-up" conformation but not for the 2nd "up" protomer in this conformation. Hence, even if the blockage of ACE2 binding in the "1-up" conformation from two independent sites may contribute significantly to the efficacy in neutralization by Sb23, the interprotomer-medicated blockage is reduced in the "2-up" conformation.

## Discussion

Currently there are no clinically validated vaccines or drugs against SARS-CoV-2 available. Developing therapeutics is currently an ongoing worldwide effort and the identification of neutralizing antibodies constitutes a key approach to that. Given the fast spreading of the disease among the population, the

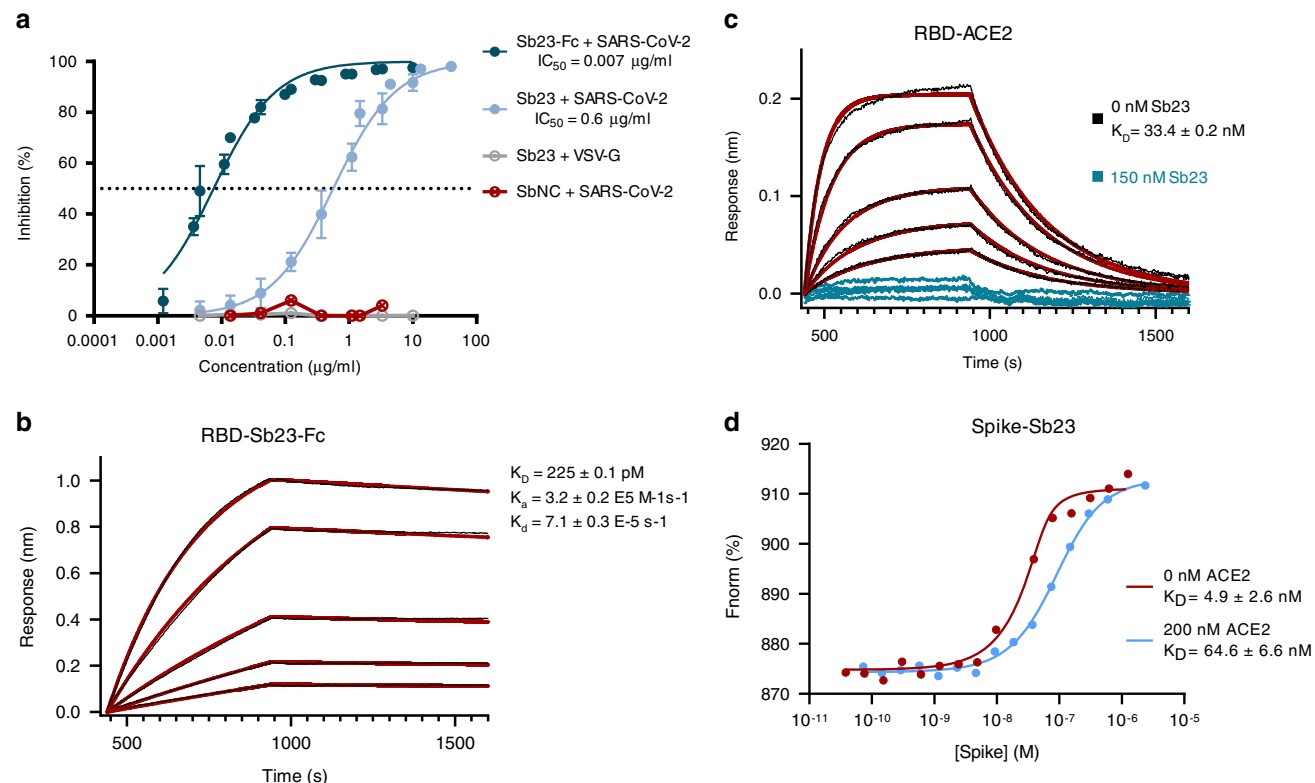

**Fig. 3 Sb23 neutralizes SARS-CoV-2 pseudoviruses and competes with ACE2. a** SARS-CoV-2 or VSV-G spike pseudotyped lentivirus was incubated with a dilution series of Sb23, Sb23-Fc, or a control sybody (specific for hPepT2). Neutralization by Sb23 is a representative of two independent experiments. Data are mean ± SD of six replicate experiments. Neutralization by Sb23-Fc represents two independent assays, performed in duplicates. Data are mean ± SD of two or four replicate experiments. **b** BLI sensorgrams of immobilized SARS-CoV-2 RBD with 2-fold serial dilution of 20 nM Sb23-Fc. Binding curves are colored black and the global fit of the data to a 1:1 binding model is red. **c** BLI sensorgrams of immobilized SARS-CoV-2 RBD with ACE2 in the presence (blue) or absence (black) of 150 nM Sb23. The assay was performed in a concentration range of 200–12.5 nM ACE2 and fit of the data to a 1:1 binding model is shown in red. **d** Microscale thermophoresis (MST) binding data of spike with fluorescently labeled Sb23, in the presence or absence of 200 nM ACE2. One representative measurement is shown. Three independent measurements were performed and affinities of spike to Sb23 in the absence of ACE2 ranged from 0.6 to 10 nM, while they were significantly lower in the presence of ACE2 ($K_D = 58$–200 nM).

research community has been trying to exploit innovative methods and platforms for the development of vaccines or neutralizing agents in the shortest possible time[32]. This methodology is an attempt to minimize the overwhelming impact that SARS-CoV-2 is having on the healthcare systems and to prepare for future pandemics. Thus, the R&D community can act in a prompter way for the development of efficient medication. Here we demonstrate that it is possible to select highly specific binders with neutralizing activity against SARS-CoV-2 from a synthetic nanobody library in a timeframe of only 2–3 weeks. The traditional generation of nanobodies requires at least 6 weeks for the Llama immunization and a total of 3–4 months for the entire selection approach[33]. The sybody platform does not require any immunization steps, has a large and diverse binder repertoire in order to account for the antibody maturation that naturally occurs in vivo and only requires a fraction of the purified antigen for the entire selection procedure[27]. The use of nanobodies as therapeutic agents has been validated by the first nanobody drug approved in 2019[34]. Synthetic libraries could boost this new technology in the establishment of platforms for the development of nanobody-based drugs, with special importance in times that fast methodologies are necessary.

In this work, we were able to identify 85 unique binders from a single selection round. None of the unique sybodies identified in this work were identical among two independent selections using

the same libraries[35,36] thus highlighting the high diversity of the three libraries and their potential for selecting several different high-affinity binders. It should be noted that different constructs of RBD and selection approaches were used for the independent studies.

Six of our analyzed sybodies (Sb12, Sb23, Sb42, Sb76, Sb95, and Sb100) bound RBD with affinities of 24.2, 10.6, 5.0, 58.1, 43.9, and 38.7 nM, respectively. The neutralization activity of the sybodies was tested using lentiviral particles pseudotyped with the SARS-CoV-2 spike protein. Sb23 was identified as the most potent neutralizer with an $IC_{50}$ of 0.6 µg/ml and 0.007 µg/ml for Sb23 and Sb23-Fc, respectively (Supplementary Table 1). Sb23 displays higher affinity for SARS-CoV-2 RBD than the human receptor ACE2 and competition assays suggest that Sb23 and ACE2 compete for the same or overlapping binding site. Based on initial models for the RBD–Sb23 complex derived from SAXS data it became clear that Sb23 binds next to the ACE2-binding site causing a steric hindrance for ACE2 to bind.

A cryo-EM reconstruction of the spike revealed that Sb23 bound two conformations of the spike, one "1-up" and one "2-up" conformation. The "2-up" conformation of the spike naturally confers extra avidity with its two binding sites for ACE2.

We can at this stage only speculate if the binding of Sb23 itself induces a conformational change that results in the "2-up" conformation or if Sb23 promotes a conformational stabilization of

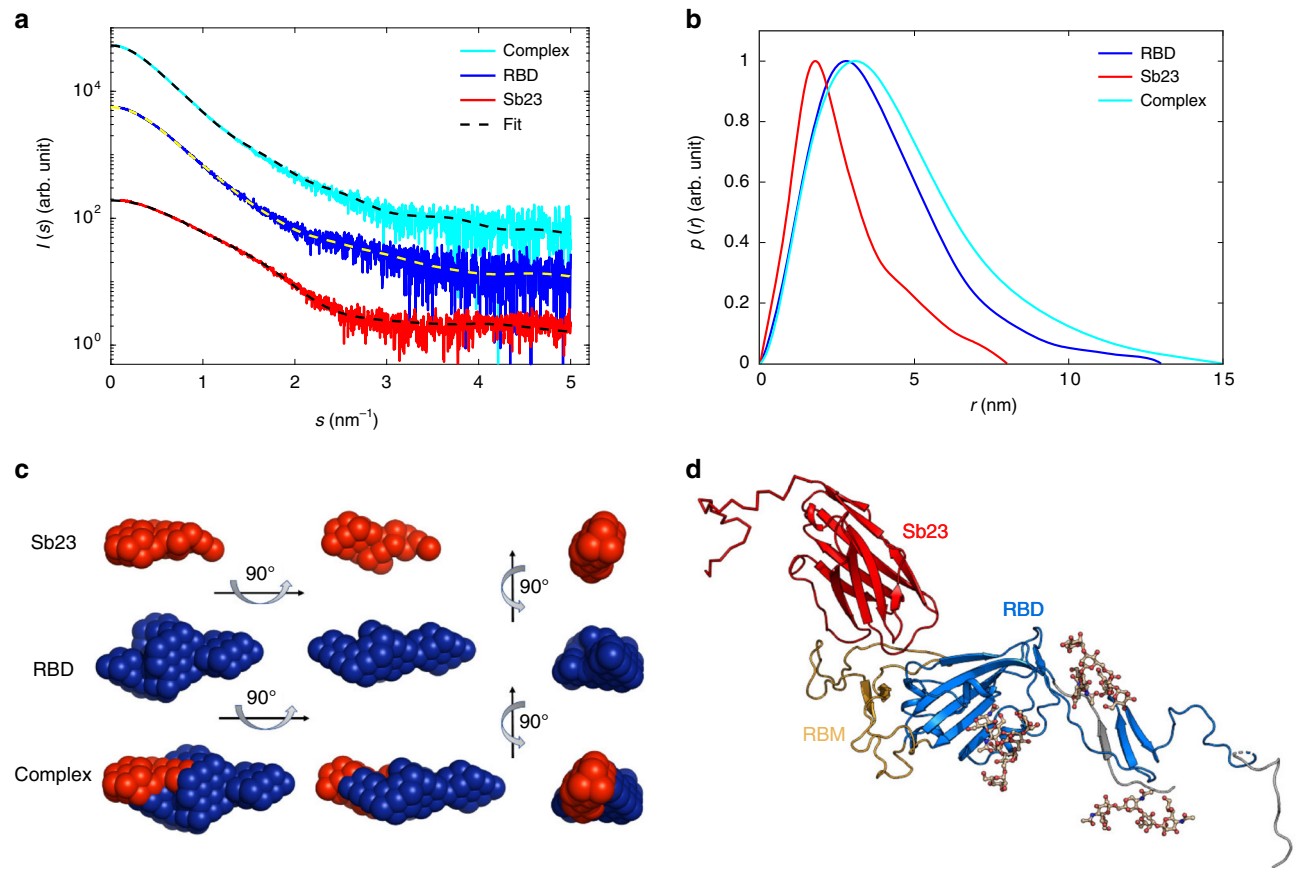

**Fig. 4 SAXS analysis of the Sb23–RBD complexes. a** Experimental SAXS data from Sb23, RBD, and its complex. **b** Distance distribution functions of Sb23, RBD, and their complex. **c** Two-phase MONSA shape of Sb23 (red beads) and RBD (blue beads). **d** Hybrid model of Sb23 in complex with RBD.

the "2-up" in the spike's dynamical landscape. Clearly, this conformation makes epitopes accessible for the development of therapeutic binders in the central cavity of the spike, this includes the lower portion of the RBD and possibly also the central helical region. Hence, the use of Sb23-bound spike protein preparations may be an excellent avenue to develop novel binders in these regions. Furthermore, the cryo-EM reconstruction of Sb23-bound spike will be crucial to develop either multimeric preparations of Sb23[37] with even increased affinity and neutralization ability or structurally based combination therapy using Sb23 in combination with other neutralizing binders of different origin.

A cryo-EM complex, which became available while this work was under review, between a traditional monoclonal FAB (FAB_C105) and SARS-CoV2-Spike shows binding to both a "2-up" and a "3-up" conformation[38]. Interestingly, FAB_C105 and Sb23 bind at two different but partially overlapping regions on the RBD (Supplementary Fig. 6). Sb23 binds on the outer RBD surface that is opposite to the FAB_C105 RBD-binding site. Thereby, Sb23 can bind the RBD in both "up" and "down" conformation. On the other hand, FAB_C105 can only bind to RBD in the "up" conformation since its binding site is not available in the RBD down conformation as FAB_C105 is sterically blocked by a neighboring FAB_C105 variable heavy chain binding an RBD in the up conformation (Supplementary Fig. 6). Arguably, this shows the strength of small high-affinity and strongly neutralizing SARS-CoV-2 Spike binders such as the Sb23 in that they are not as often sterically occluded in the same fashion as traditional much larger binders such as FABs.

Furthermore, among the sybodies that displayed IC$_{50}$ values below 20 µg/ml, we have identified a group of sybodies (Sb12, 76,

and 100) that can bind simultaneously with Sb23 to RBD and were also able to abolish or decrease the binding affinity of ACE2 (Supplementary Fig. 7). For the development of therapeutic agents, it would be advantageous to further increase the avidity and affinity of these binders. Heterobivalent ligands are well known in the pharmacology area, for their distinctly increased avidity and residence time over the individual ligands[39]. By combining the neutralization potential of Sb23 with a non-overlapping sybody, we could potentially increase the overall avidity towards the spike protein. Indeed, the bispecific chimeric protein created by fusing Sb23 with Sb12 (Sb23/Sb12) displays neutralization potential and affinity to RBD in the same range as found for Sb23-Fc or Sb42-Fc (Supplementary Fig. 7g, h), highlighting the strength of such heterobivalent ligands.

The COVID-19 pandemic illustrates how previously unknown viral strains can emerge and quickly spread across continents, causing hundreds of thousands of fatalities in a very short time period. The rapid fabrication of treatments against new viral strains is a key component to prevent such fatality rates in the future. Synthetic libraries are an alternative approach to rapid drug development, quickly generating highly specific binders with neutralization potential.

## Methods

**Construct design, expression, and purification of SARS-CoV-2 proteins.** The plasmid for the expression of the SARS-Cov2 prefusion-stabilized spike protein was obtained by[10]. In short, the gene encodes residues 1–1208 of 2019-nCoV S (GenBank: MN908947) with proline substitutions at residues 986 and 987, a "GSAS" substitution at the furin cleavage site (residues 682–685), a C-terminal T4 fibritin trimerization motif, a HRV3C protease cleavage site, a Twin-Strep tag, an 8 × Histidine tag and a secretion signal. The gene for RBD expression comprises

**Table 1 Data collection and structure statistics for SAXS analysis.**

| Data collection parameters | Sb23 | RBD | Sb23+RBD |
|---|---|---|---|
| Data collection parameters | | | |
| Instrument | | EMBL P12 (PETRA III, DESY, Hamburg) | |
| Beam geometry (mm²) | | 0.2 × 0.12 | |
| Wavelength (nm) | | 0.124 | |
| s range (nm⁻¹) | | 0.03–5.0 | |
| Exposure time (s) | | 4 (20 × 0.2 s) | |
| Temperature (K) | | 293 | |
| Concentration range (mg ml⁻¹) | | 0.37–4.0 | |
| Structural parameters | | | |
| $R_g$ (nm) (from $P(r)$) | 2.2 ± 0.1 | 3.2 ± 0.2 | 3.7 ± 0.2 |
| $R_g$ (nm) (from Guinier plot) | 2.1 ± 0.1 | 3.1 ± 0.2 | 3.5 ± 0.2 |
| $D_{max}$ (nm) | 8.0 ± 0.5 | 13 ± 1 | 15 ± 1 |
| Porod volume estimate, $V_p$ (nm³) | 20 ± 2 | 66 ± 2 | 86 ± 5 |
| Molecular weight determination (kDa) | | | |
| From Porod volume ($V_p/\sim$1.6) | 13 ± 1 | 37 ± 5 | 54 ± 3 |
| From consensus Bayesian assessment | 15 ± 3 | 41 ± 1 | 53 ± 6 |
| From $I(0)$ | 21 ± 2 | 33 ± 9 | 62 ± 9 |
| Calculated monomeric $MW$ from sequence | 15.7 | 32.2 | 47.9 |
| Software employed | | | |
| Primary data reduction | SASFLOW | SASFLOW | SASFLOW |
| Data processing | PRIMUS | PRIMUS | PRIMUS |
| Rigid body modeling | CORAL | CORAL | SASREF |
| Computation of model intensities | CRYSOL | CRYSOL | CRYSOL |
| 3D graphics representations | PYMOL | PYMOL | PYMOL |

residues (319–566) of 2019-nCoV RBD-SD1 and was cloned into a paH mammalian expression vector containing a secretion signal, a C-terminal Sortase recognition motif, and a non-cleavable Histidine tag[25]. Residues 1–616 of human ACE2 including the native N-terminal secretion sequence (residues 1–17) were cloned in a pXLG vector with a C-terminal HRV3C protease cleavage site and 2x FLAG Flag tag for purification.

Suspension adapted HEK293-F cells were grown in 600-ml Bioreactors in 250 ml of serum-free FreeStyle medium in an incubator at 37 °C with humidified atmosphere at 8% $CO_2$ on an orbital shaker platform rotating at 220 rpm. Cells were grown to a density of $0.4$–$3 \times 10^6$ cells/ml. Cell counts and viabilities were determined by using standard Trypan blue exclusion methods in cell counter (BioRad).

The spike and RBD constructs were transfected and expressed according to the following protocol[40]. One day before transfection, cells were seeded in culture medium at a density of $2 \times 10^6$ cells/ml. On the day of transfection, cells were centrifuged (5 min, RT, $100 \times g$) and resuspended in fresh FreeStyle medium to a final density of $20 \times 10^6$ cells/ml. DNA was added to a final concentration of 1.5 mg/l, mixed and followed by addition of MAX PEI in 1:2 ratio (w/w). Transfected cells were incubated at 37 °C with agitation at 220 rpm in an 8% $CO_2$ atmosphere for 45 min. After this incubation time, fresh FreeStyle medium was added to reach a final cell density of $1 \times 10^6$ cells/ml. Media with secreted proteins were harvested 4 days post-transfection ($10,000 \times g$, 20 min, 4 °C) and filtered (pore size of 0.22 μm) prior purification.

For the ACE2 transfection, DNA was added to a final concentration of 0.75 μg/$10^6$ cells, mixed and followed by the addition of Linear PEI 25 K in a 1:2 ratio (w/w). The expression was conducted in glass flasks with an initial cell concentration of $2 \times 10^6$ cells/ml in the presence of 2 mM sodium butyrate. Media containing the secreted ACE2 protein was harvested 2 days post-transfection ($1000 \times g$, 20 min, 4 °C).

SARS-CoV-2 prefusion spike for sybody selection, biophysical characterization, and cryo-EM was expressed and purified as previously described[25]. For the purification of the RBD domain, IMAC affinity chromatography was used[41]. To 1 l of filtered medium imidazole was added to a final concentration of 15 mM. Medium was loaded onto two 5 ml HisTrap™ High-performance columns equilibrated with wash buffer (20 mM Na-P pH 7.4, 300 mM NaCl, 15 mM imidazole), connected in tandem to the ÄKTAPure chromatography system using a sample pump at a flow rate of 2 ml/min. Columns were washed with wash buffer (20 mM Na-P pH 7.4, 300 mM NaCl, 15 mM imidazole) until the absorbance

reached a steady baseline. RBD was eluted in one-step with elution buffer (20 mM Na-P pH 7.4, 150 mM NaCl, 300 mM imidazole). Eluted protein was concentrated by ultrafiltration using Corning® Spin-X® UF concentrators with a cut-off limit of 10 kDa prior gel filtration. The concentrated protein sample was loaded on a Superdex 200 Increase 10/300 GL column (GE Healthcare) equilibrated with GF buffer (50 mM TRIS pH 7.4, 150 mM NaCl) using an ÄKTAPure chromatography system. The elution profile was monitored at 280 nm and collected fractions containing the target protein were concentrated to approximately 10 mg/ml, flash frozen in liquid nitrogen and stored at −80 °C until further use.

For the ACE2 purification, the cell culture medium was mixed with 10× washing buffer (500 mM TRIS pH 8.0, 1.5 M NaCl) and loaded onto M2 FLAG beads. The beads were extensively washed with wash buffer (50 mM TRIS pH 8.0, 150 mM NaCl) and eluted with 3xFLAG peptide. The protein was further polished by gel filtration on a Superdex 200 10/300 GL column in 50 mM HEPES pH 7.2, 150 mM NaCl, 10% glycerol, concentrated and flash frozen in liquid nitrogen until further use.

All proteins were subject to quality control and initial biophysical characterization (Supplementary Fig. 8).

**Chemical biotinylation**. The RBD and spike proteins were chemically biotinylated using the EZ-Link NHS-PEG4-Biotin kit (Sigma). For the RBD, the sample was first desalted using a PD-10 gravity flow column (GE Healthcare) in 20 mM Na-P pH 7.4 and 150 mM NaCl. Subsequently, the sample was chemically biotinylated for 2 h on ice, using a 20-fold molar excess of biotin over the target protein. The spike sample was chemically biotinylated in a similar manner, using a 10-fold molar excess of biotin for 1 h on ice. Both samples were further run on gel filtration, following the protocol described above to remove non-reacted NHS-PEG4-Biotin. The fractions containing the target protein were collected and concentrated to approximately 1 mg/ml. 5% glycerol was added before flash-freezing, and the samples were stored at −80 °C until further use.

**Sybody selections, and ELISA**. Sybody selections and ELISA were performed as described in detail[26,27]. Sybody selections against biotinylated RBD were carried out using three synthetic libraries diverging in the length of CDR3 loop, dubbed concave, loop, and convex[26]. One round of ribosome display followed by two rounds of phage display were performed. All rounds were monitored using qPCR. To obtain binders with increased affinities, one off-rate selection was performed during the second round of phage display, using 5 μM of non-biotinylated RBD with a 5 min incubation time. Biotinylated maltose-binding protein (MBP) was used as a negative control to determine the enrichment factor for each phage display round. 94 individual colonies were randomly selected from each library and cultured in TB medium for ELISA. The sybodies were expressed by inducing with 0.02% (w/v) L-Arabinose and the positive clones expressing RBD-specific sybodies were identified by performing a periplasmic extraction and analyzed against the two target proteins, biotinylated RBD and biotinylated spike. The same procedure was performed for the MBP, used as background control signal.

Periplasmic extracts were transferred to microtiter plates, previously coated with Protein A (Sigma), blocked in TBS-BSA buffer, and subsequently incubated with anti-myc antibody (Sigma). After capturing the individual sybodies, biotinylated target and control proteins were added at 50 nM concentration. The last incubation step was performed by adding streptavidin-HRP (Sigma). Between incubation steps, the plates were washed three times in TBS-BSA containing 0.05% Tween-20. ELISA signals were developed by adding TMB (Sigma) in 50 mM $Na_2HPO_4$, 25 mM citric acid and 0.006% $H_2O_2$. ELISA signals were measured at an absorbance of 650 nm.

**Cloning, expression, and purification of sybodies**. The identified sybody genes of positive clones were chemically transformed into *Escherichia coli* MC1061 F- (Lucigen) cells. The cells were cultured in TB medium supplemented with chloramphenicol (34 μg/mL) at 37 °C shaking at 200 rpm. Once the optical density reached 0.4–0.8 (at 600 nm), the cultures were induced with the addition of 0.02% (w/v) L-Arabinose and incubated at 22 °C, 200 rpm overnight. The proteins in the periplasmic extract were released by osmotic shock and purified by immobilized metal affinity chromatography (IMAC) using Ni-NTA resin (Invitrogen) in TBS buffer (20 mM Tris, 150 mM NaCl, pH 7.4). The purified sybodies were desalted in PBS (10 mM phosphate, 2.7 mM KCl and 137 M NaCl, pH 7.4) buffer, analyzed by SDS-PAGE and gel filtration using a SRT SEC-100 (Sepax Technologies) column. For heterobivalent binding, Sb23 was fused with Sb12 via a G-S linker with a length of 12 residues. The chimeric construct was expressed and purified in the same manner as the other sybodies.

Sb23 or Sb42 were cloned into pCMVExt-Fc[42] for mammalian expression, just before the Fc domain from Human IgG. Transfection and expression of the Sb-Fc constructs were performed as described before. The protein was purified from the filtered media by immobilized metal affinity chromatography (IMAC) using Ni-NTA resin (Invitrogen) in TBS buffer and desalted in PBS buffer. All primer sequences used for preparation of the Sb or Fc fusion constructs can be found in Supplementary Table 2.

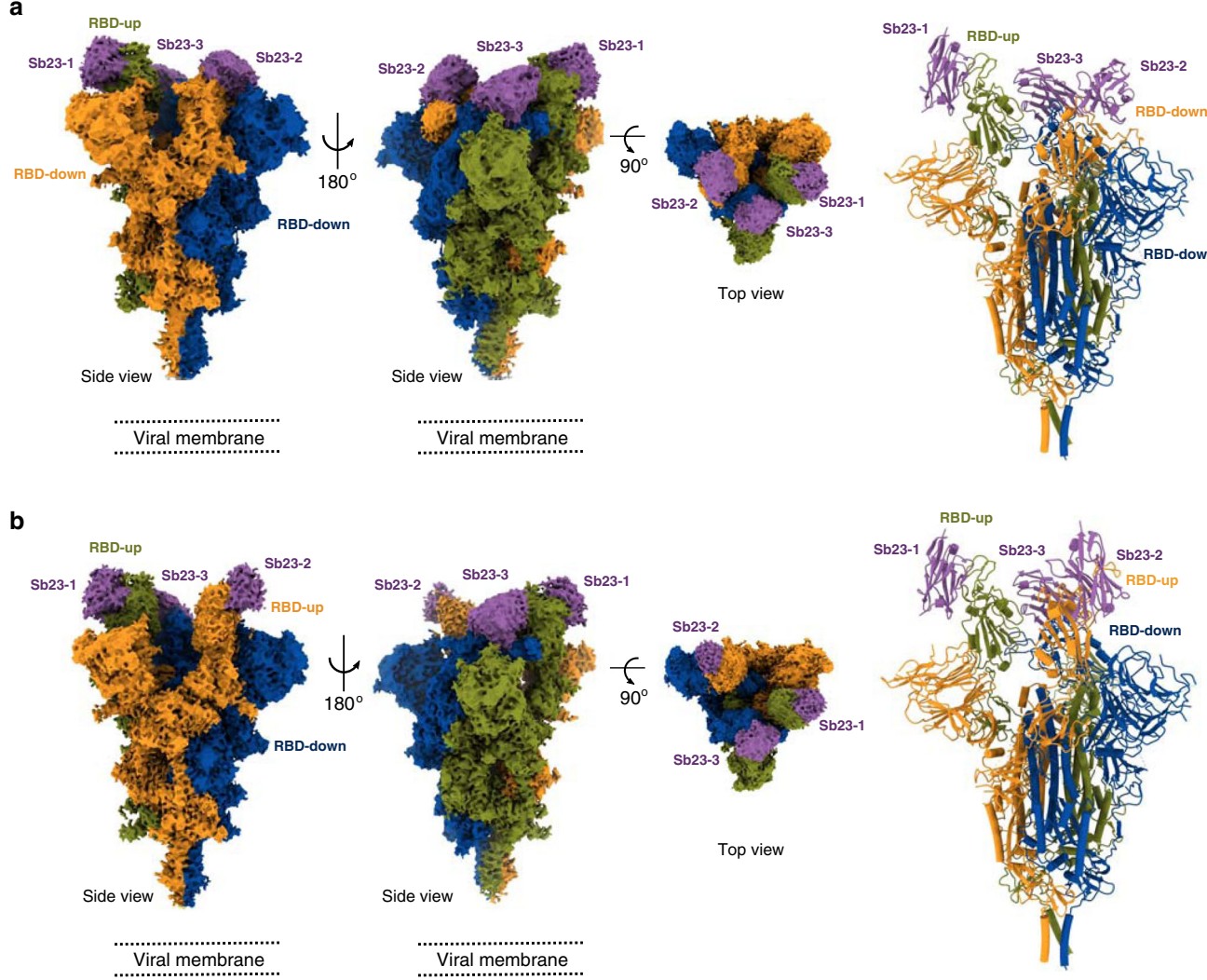

**Fig. 5 Cryo-EM reconstruction of SARS-CoV-2 spike bound to Sb23. a** Locally sharpened Coulomb potential map and cartoon model of Sb23 bound to the spike protein in the "1-up" conformation and cartoon model of Sb23-bound spike. **b** Locally sharpened Coulomb potential map and cartoon model of Sb23 bound to the spike protein in the "2-up" conformation.

**Neutralization assay**. Pseudotyped neutralization assays were adapted from protocols previously validated to characterize the neutralization of HIV[43], but with the use of HEK293T-ACE2 cells as previously described[25]. Briefly, pseudotyped lentiviruses displaying the SARS-CoV-2 spike protein (harboring an 18 amino acid truncation of the cytoplasmic tail[44]) and packaging a luciferase reporter gene were generated by the co-transfection of HEK293T cells. Media was changed 12–16 h after transfection, and pseudotyped viruses were harvested at 48- and 72-h post-transfection, filtered through a 0.45 μm filter, and stored at −80 °C until use. Pseudotyped viruses sufficient to generate ~100,000 RLUs were incubated with serial dilutions of sybodies for 60 min at 37 °C, and then ~15,000 HEK293T-ACE2 cells were added to each well. Plates were incubated at 37 °C for 48 h, and luminescence was measured using Bright-Glo (Promega) on a GM-2000 luminometer (Promega).

**Thermal stability assays**. The thermal stability of RBD, ACE2, and spike was monitored by nanoDSF (NanoTemper Technologies) at a concentration of 0.25 mg/ml in PBS (10 mM phosphate, 2.7 mM KCl and 137 M NaCl, pH 7.4) buffer. To monitor binding-induced thermal shift changes of the RBD with different sybodies, RBD at 0.25 mg/ml was incubated with 0.15 mg/ml of different sybodies (Sb23, Sb76, Sb95, control Sb) and incubated for at least 10 min at RT prior analysis. Standard grade nanoDSF capillaries (NanoTemper Technologies) were loaded into a Prometheus NT.48 device (NanoTemper Technologies) controlled by PR. ThermControl (version 2.1.2). Excitation power was adjusted to 30% and samples were heated from 20 °C to 90 °C with a slope of 1 °C/min. All samples were run in triplicates and error bars represent standard deviations. The stability of

the different sybodies was analyzed in independent experiments and they all display transition temperatures in the range of 70–95 °C.

**Far-UV CD spectroscopy**. RBD, spike protein, and ACE2 were dialyzed or diluted by the buffer containing 10 mM Na-P (pH 7.5) and 20 mM NaCl to 0.11, 0.13, and 0.08 mg/ml, respectively. The CD spectrum was measured on a Chirascan CD spectrometer (Applied Photophysics) in a 0.1 mm cuvette between 195 and 260 nm with 1 nm steps ten times. The resulting spectra were averaged, buffer-subtracted and the secondary structure was estimated by the K2D algorithm using Dichroweb[45].

**Biolayer interferometry (BLI)**. The binding of selected sybodies to RBD or binding of RBD to ACE2 proteins was measured by biolayer interferometry (BLI) using the Octet RED96 system (FortéBio).

For the RBD-binding screening assay, biotinylated RBD was loaded on streptavidin biosensors (FortéBio), pre-equilibrated in assay buffer (PBS-buffer supplemented with 0.5% (w/v) BSA and 0.05% (v/v) Tween-20) for 20 min. Prior to association, a baseline step of 60 s was performed. Subsequently, sensors were dipped in a well containing 500 nM target sybody (association step) for 10 min followed by 15 min of dissociation time in the same buffer.

Concentration-dependent kinetic assays were performed in a similar manner, by dipping RBD-captured streptavidin biosensors in different concentrations of target protein (sybodies 23, 42, 76, and 95 or ACE2) ranging from 5 to 250 nM for 8 min followed by a 12 min dissociation step. Competition assays of RBD-ACE2 with Sb23 were performed in an identical manner, containing 150 nM final concentration of Sb23 in all wells. All experiments were carried out at 22 °C.

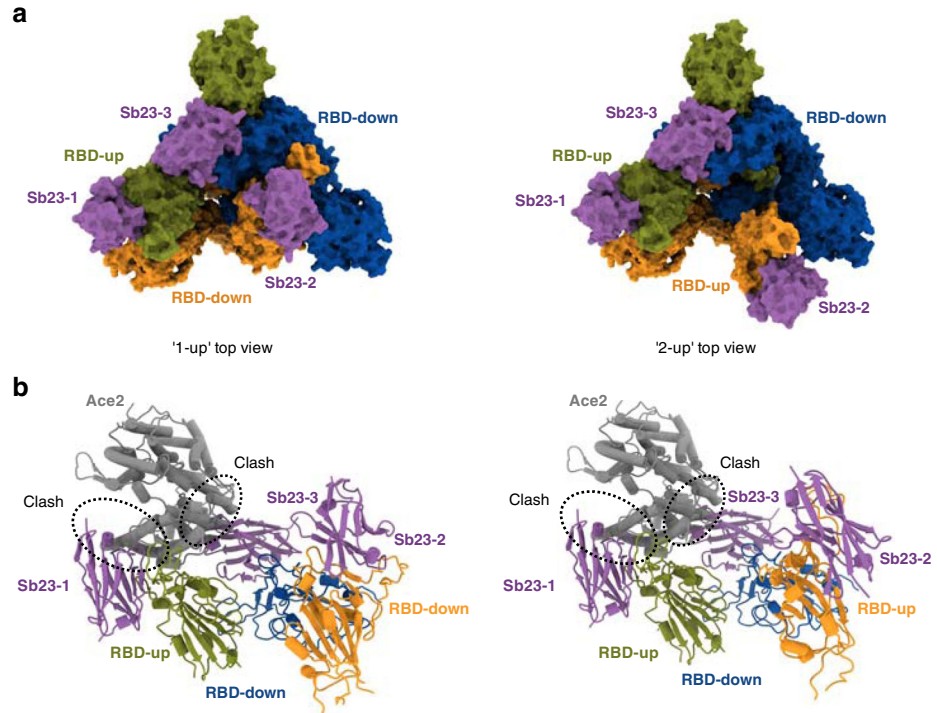

**Fig. 6 Top view of cryo-EM reconstruction of SARS-CoV-2 spike bound to Sb23 and modeling of the structural basis for Sb23-based blockage of SARS-CoV-2 spike binding to ACE2. a** Top view of locally sharpened Coulomb potential map and cartoon model of Sb23 bound to the spike protein in the "1-up" conformation **b** Top view of locally sharpened Coulomb potential map and cartoon model of Sb23 bound to the spike protein in the "2-up" conformation. **c** Cartoon model of Sb23-bound Spike in the "1-up" (left) "2-up" (right) conformation showing how ACE2 binding is blocked by Sb23 bound to the RBD in the "up" conformation as well as Sb23 bound to the neighboring RBD in the down conformation.

Data were reference-subtracted and aligned with each other in the Octet Data Analysis software v10.0 (FortéBio), using a 1:1 binding model. All figures were prepared using Prism 8 (GraphPad).

**Microscale thermophoresis (MST)**. The binding affinity of spike to Sb23 was measured by microscale thermophoresis using the MST NT.115 device (NanoTemper Technologies). Sb23 was labeled via the His₆-tag with RED-tris-NTA dye (Nano-Temper Technologies) in PBS-T (10 mM phosphate, 2.7 mM KCl and 137 M NaCl, pH 7.4 supplemented with 0.05% (v/v) Tween-20), following the manufacturer's instructions. A 2-fold dilution series of spike protein was prepared in PBS-T with the highest concentration in the assay being 1250 nM. A 50 nM final concentration of labeled Sb23 was mixed with the substrate dilution series and incubated for 10 min at room temperature before loading the samples in standard capillaries. For the competition assay, the same procedure was used but the spike dilution series were prepared in PBS-T buffer containing 800 nM of ACE2 (200 nM final concentration in the assay) and incubation of 10 min prior to mixing with labeled Sb23. All measurements were performed in triplicate using 35% LED and medium MST power. The data were analyzed using the MO. Affinity Analysis software (NanoTemper Technologies). Figures were prepared using Prism 8 (GraphPad).

**Small angle X-ray scattering measurements (SAXS)**. The SAXS data were collected at the EMBL P12 beamline of the storage ring PETRA III (DESY, Hamburg, Germany) using a robotic sample changer[46]. The scattering from the protein solutions and respective solvents was recorded on a Pilatus-6M pixel detector, radially averaged and processed using standard procedures. For all constructs, a concentration series was measured in the range of about 0.3–4 mg/ml and the data were extrapolated to infinite dilution as needed. The radial average was performed and solvent scattering was subtracted using the SASFLOW pipeline[47]. The distance distributions were computed by GNOM[48] and the overall parameters were determined from the reduced data using relevant programs from the ATSAS suite[47]. The ab initio low-resolution shapes were restored using the program MONSA[49]. Scattering from the high-resolution models was computed by CRY-SOL[50]. Hybrid models of the sybodies and of the RBD were constructed by CORAL[51] using the available high-resolution portions as rigid bodies and amending them by chains of dummy residues to represent the fragments not present in the crystal. For RBD, three glycans were added to the most frequent relative abundance of glycan species, the C-terminal portion was modeled as a coil, and the structure was refined using SREFLEX[52]. The model of Sb23–RBD complex was constructed using the rigid body modeling program SASREF[53].

**Cryo-EM sample preparation and Imaging**. Spike trimer (0.7 mg/ml) and Sb23 (1.3 mg/ml) were mixed in a 1:3 molar ratio and incubated on ice for 5 min. A 3-μl aliquot of the sample solution was applied to glow-discharged CryoMatrix holey grids with amorphous alloy film (Zhenjiang Lehua Technology) in a Vitrobot Mk IV (Thermo Fisher Scientific) at 4 °C and 100% humidity (blot 10 s, blot force 3). Cryo-EM data collection was performed with EPU 2.7 (Thermo Fisher Scientific) using a Krios G3i transmission-electron microscope (Thermo Fisher Scientific) operated at 300 kV in the Karolinska Institutet 3D-EM facility. Images were acquired in nanoprobe EFTEM mode with a slit width of 10 eV using a Bio-Quantum K3 (Ametek) during 1.5 s with a flux of 8.7 e⁻/px/s resulting in a fluence of 50 e⁻/Å² equally fractionated into 60 frames. Motion correction, CTF-estimation, Fourier binning (to 1.02 Å/px), particle picking, and extraction were performed on the fly using Warp[54]. A total of 8,236 micrographs that fulfilled the required criteria (defocus less than 2 microns and estimated resolution better than 4 Å) were selected and 387,116 particles were picked by Warp. Extracted particles were imported into cryoSPARC v2.15.0[55] for 2D classification, heterogenous refinement, and non-uniform 3D refinement[56]. The particles were processed with C1 symmetry throughout. After 2D classification (300 classes) 199,501 particles were retained and used to build three ab initio 3D reconstructions. These were further processed for heterogeneous refinement that resulted in one reconstruction showing high-resolution structural features in the core of the spike. After one round of homogenous refinement we performed 3D variability analysis using three orthogonal principle modes. In the ensuing analysis, we separated out four different clusters of which two gave clusters that showed clearly contrasting reconstructions after separate homogenous reconstructions ("1-up" and "2-up"). These two reconstructions were then used together with two "crap" classes in high-resolution heterogeneous refinement of the original particle set selected from 2D classification. The overall resolution of the "1-up" and "2-up" conformations using non-uniform refinement[56] is 3.1 Å and 2.9 Å (0.143 FSC) using 70,854 and 69,567 particles, respectively. The local resolution of the density in the very top of the spike is lower (Supplementary Fig. 5). All processing and local resolution plotted on the electron density surface are shown in Supplementary Fig. 5 and reconstruction and refinement details are found in Table 2.

**Model building and structure refinement**. A Cryo-EM structure of the SARS-CoV-2 spike protein trimer[10] (PDB: 6VSB) was used as a starting model for model building. The model was extended and manually adjusted in COOT[57]. The sybody structure was homology modeled using Phyre2[58] with PDB: 4PFE[59] as a template. The missing regions of the RBD domains were built based on the RBD-spike crystal structure (PDB: 6LZG)[31]. Structure refinement and manual model building were

**Table 2 Cryo-EM data collection, refinement, and validation statistics.**

| | #1 SPike-SB23 1-up (EMDB-11616) (PDB 7A25) | #2 SPIKE-SB23 2-UP (EMDB-11617) (PDB 7A29) |
|---|---|---|
| Data collection and processing | | |
| Magnification | 165,000x | – |
| Voltage (kV) | 300 kV | – |
| Electron exposure (e⁻/Å²) | 50 | – |
| Defocus range (μm). | 0.3–1.1 | – |
| Pixel size (Å). | 0.51 Å | – |
| Symmetry imposed | C1 | – |
| Initial particle images (no.) | 387,116 | – |
| Final particle images (no.) | 70,854 | 69,567 |
| Map resolution (Å) | 3.1 | 2.9 |
| FSC threshold | 0.143 | 0.143 |
| Map resolution range (Å) | 2.1–7.5 Å (FSC 0.143) | 2.1–7.2 Å (FSC 0.143) |
| Refinement | | |
| Initial model used (PDB code) | 6VSB, 6LZG, 4PFE | 6VSB, 6LZG, 4PFE |
| Model resolution (Å) | 3.1 Å | 2.9 Å |
| FSC threshold | 0.5 | 0.5 |
| Model resolution range (Å) | 230–3.1 | 250–2.9 |
| Map sharpening $B$ factor (Å²) | −57 | −62 |
| Model composition | | |
| Non-hydrogen atoms | 28266 | 28286 |
| Protein residues | 3504 | 3510 |
| Ligands | 63 (NAG) | 61 (NAG) |
| $B$ factors (Å²) | | |
| Protein | 82 | 115 |
| Ligand | 107 | 125 |
| R.m.s. deviations | | |
| Bond lengths (Å) | 0.003 | 0.002 |
| Bond angles (°) | 0.672 | 0.531 |
| Validation | | |
| MolProbity score | 1.96 | 1.68 |
| Clashscore | 7.85 | 7.95 |
| Poor rotamers (%) | 2.08 | 0.03 |
| Ramachandran plot | | |
| Favored (%) | 95.76 | 96.29 |
| Allowed (%) | 4.24 | 3.71 |
| Disallowed (%) | 0.00 | 0.00 |

performed using COOT and PHENIX[60] in interspersed cycles with secondary structure and geometry restrained. All structure figures and all EM density-map figures were generated with UCSF ChimeraX[61].

**Reporting summary**. Further information on research design is available in the Nature Research Reporting Summary linked to this article.

## Data availability

The scattering data and models of Sb23, RBD and their complex are deposited into SASBDB (www.sasbdb.org), entries SASDJF4, SASDJG4 and SASDJH4 respectively. The MST and BLI data are available under https://github.com/tania-custodio/Sb23. The cryo-EM density maps of SARS-CoV-2 spike glycoprotein with Sb23 bound were deposited in the Electron Microscopy Data Bank (EMDB) with accession codes EMD-11616 (1-up) and EMD-11617 (2-up). The corresponding models were deposited in the Protein Data Bank (PDB) with accession codes 7A25 (1-up) and 7A29 (2-up). The sequences of all the selected sybodies from this study are provided in Supplementary Data 1. Any other experimental data that support the findings of this study are available from the corresponding authors upon request. Source data are provided with this paper.

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

## Acknowledgements

The authors thank the Sample Preparation and Characterization facility at EMBL (Hamburg, Germany) for their support with nanoDSF, MST, and biolayer interferometry measurements. We thank Stephan Niebling for fruitful advice with biolayer interferometry analysis, Jason McLellan for prefusion spike plasmid and all group members for discussions. The authors thank James Voss and Deli Huang for neutralization assay reagents. All cryo-EM data were collected at the Karolinska Institutet's 3D-EM facility. This work was supported in part by an EU H2020 grant (CoroNAb) to BM and GM, by project grants from the Swedish Research Council to BM (2018-02381), BMH (2017-6702 and 2018-3808), GM (2018-03843) and the Knut and Alice Wallenberg Foundation to BMH.

## Author contributions

T.F.C. and J.P. selected sybodies under the supervision of M.S. and M.A.S., M.S. and M.A.S. contributed reagents to the project. T.F.C., L.H., S.P., J.P., N.D., and K.R. expressed and purified SARS-CoV-2 proteins and sybodies. T.F.C., J.P., S.P., and C.L. characterized proteins and their interactions in vitro. D.J.S. performed neutralization assays under the supervision of B.M., M.S., A.G., C.J., M.G., and D.S. collected and analyzed SAXS data. H.D. and B.M.H. developed vitrification conditions, collected cryo-EM data and determined the structure. The project was conceived and supervised by G.M.M., B.M., B.M.H. and C.L. C.L., T.F.C., S.P., and B.M.H. wrote the first draft of the manuscript and received input from all other authors.

## Funding

## Competing interests

The authors declare no competing interests.
