## [Peer Review File · Nature Communications]

Reviewers' Comments:

Reviewer #1:

Remarks to the Author:

In this paper the consortium retrieved multiple synthetic nanobodies from the Sy library, of which one (Sb23) was characterised in more detail (affinity, competition with ACE2, target stabilisation, SAXS, cryo-EM and virus neutralisation). The cryo EM is interesting as it shows a new conformation of its three RBDs.

In this report again, the Sy library seems to confirm its value to generate potent affinity reagents.

The experiments are properly performed, the data are well analysed and interesting, and accurately explained and discussed.

However, I would like to take this opportunity to suggest two additions:

1. The reference 34 of Walter et al in BioRxiv is incomplete. Please add the doi
2. More importantly, the BLI sensorgrams for S23 should be discussed more. The 100 and 50 nM concentration gives nearly overlapping profiles. This suggests that 50 nM is already saturating the immobilised target. So perhaps the 100 nM tracing should be omitted. Furthermore, it is awkward to see that the plateau levels obtained for the diluted series are not spaced evenly (as for example with Sb42). So what is the reason? could it be that the binding kinetics at lower concentration and at higher concentrations are different due to cooperative effects? What I don't understand is that the red and black traces (calculated and experimental traces) are overlapping so well?

Reviewer #2:

Remarks to the Author:

Custodio et al. present a sybody ensemble targeting the spike trimer complex of SARS CoV-2. They present a procedure for isolation of inactivating hits identified on the basis of a lenti pseudovirus infection model. SAXS, other biophysical studies and single-particle cryo-EM are employed to characterize the complex and binding interactions and blocking mechanism on the basis of known features of e.g. up/down conformational changes in the spike trimer, which are important for ACE2 receptor interactions. The experiments are described OK and should be improved well by another round of revision with care to detail.

Major points

- 1) What is the clear message of the paper? This is not really clear and should be emphasised better. The isolation of sybodies has been reported earlier, and so has the description of structures of antibody complexes to the spike proteins, so the fast procedure and its application must be the particularly novel aspect. This must be documented - what are the time-reducing steps of a sybody strategy compared to other approaches, e.g. nanobodies or regular immunisation? Realistic timelines/scenarios should be discussed of how this procedure may be advantageous compared to other strategies and make a particular point of clinical relevance.
- 2) how do binding sites compare to the upcoming report of the Bjorkman group mapping several human antibodies to the spike? [https://www.cell.com/cell/pdf/S0092-8674\(20\)30757-1.pdf](https://www.cell.com/cell/pdf/S0092-8674(20)30757-1.pdf)
- 2) What would be a next step of relevance to make use of the study?
- 3) what is the important result of the SAXS data? It is described at length, but it is not obvious what it adds to the study.
- 3) Up-down conformations of the spike trimer are discussed quite at length but not illustrated well.

minor point:

4) the sequence information was missing in the supplementary material - many blank pages are provided.

REVIEWER COMMENTS

Reviewer #1 (Remarks to the Author):

In this paper the consortium retrieved multiple synthetic nanobodies from the Sy library, of which one (Sb23) was characterised in more detail (affinity, competition with ACE2, target stabilisation, SAXS, cryo-EM and virus neutralisation). The cryo EM is interesting as it shows a new conformation of its three RBDs.

In this report again, the Sy library seems to confirm its value to generate potent affinity reagents.

The experiments are properly performed, the data are well analysed and interesting, and accurately explained and discussed.

However, I would like to take this opportunity to suggest two additions:

1. The reference 34 of Walter et al in BioRxiv is incomplete. Please add the doi

We thank the reviewer for spotting the mistake. It has been corrected.

Line 566: 36. Walter, J. D. et al. Sybodies targeting the SARS-CoV-2 receptor-binding domain. bioRxiv (2020) doi:10.1101/2020.04.16.045419.

2. More importantly, the BLI sensorgrams for S23 should be discussed more. The 100 and 50 nM concentration gives nearly overlapping profiles. This suggests that 50 nM is already saturating the immobilised target. So perhaps the 100 nM tracing should be omitted. Furthermore, it is awkward to see that the plateau levels obtained for the diluted series are not spaced evenly (as for example with Sb42). So what is the reason? could it be that the binding kinetics at lower concentration and at higher concentrations are different due to cooperative effects? What I don't understand is that the red and black traces (calculated and experimental traces) are overlapping so well?

We agree with the reviewer that the sensorgram profile deviates slightly from a standard serial dilution profile, as seen for other sybodies. However, we do not have any evidence that the effect seen is due to cooperative effects, but rather from being close to saturating conditions. Starting from a lower sample concentration, will result in a more evenly spaced profile. Due to the small size of the sybodies, using lower concentrations will result in weaker signals, which in turn leads to an increase in noise and bias processing. We choose the stated concentrations to ensure a good signal to noise ratio. The figure legend has been adjusted.

Regarding the fitting procedure, the data were processed using a local fitting approach. Although global fitting is preferred over local fitting, the former results in a better fit of the data over slow-off rate profiles. The kinetic values were analysed by both local and global fitting and we obtained very similar results, for all analysed sybodies.

Reviewer #2 (Remarks to the Author):

Custodio et al. present a sybody ensemble targeting the spike trimer complex of SARS CoV-2. They present a procedure for isolation of inactivating hits identified on the basis of a lenti pseudovirus infection model. SAXS, other biophysical studies and single-particle cryo-EM are employed to characterize the complex and binding interactions and blocking mechanism on the basis of known features of e.g. up/down conformational changes in the spike trimer, which are important for ACE2 receptor interactions. The experiments are described OK and should be improved well by another round of revision with care to detail.

Major points

1) What is the clear message of the paper? This is not really clear and should be emphasised better. The isolation of sybodies has been reported earlier, and so has the description of structures of antibody complexes to the spike proteins, so the fast procedure and its application must be the particularly novel aspect. This must be documented - what are the time-reducing steps of a sybody strategy compared to other, approaches, e.g. nanobodies or regular immunisation? Realistic timelines/scenarios should be discussed of how this procedure may be advantageous compared to other strategies and make a particular point of clinical relevance.

We would like to start by thanking the reviewer for the suggestions. We have extended the discussion and have tried to emphasise the message of the paper stronger.

You now can read from line 192:

“Currently there are no clinically validated vaccines or drugs against SARS-CoV-2 available. Developing therapeutics is currently an ongoing worldwide effort and the identification of neutralizing antibodies constitutes a key approach to that. Given the fast spreading of the disease among the population, the research community has been trying to exploit innovative methods and platforms for the development of vaccines or neutralizing agents in the shortest possible time [32]. This methodology is an attempt to minimize the overwhelming impact that SARS-CoV-2 is having on the healthcare systems and to prepare for future pandemics. Thus, the R&D community can act in a prompter way for the development of efficient medication. Here we demonstrate that it is possible to select highly specific binders with neutralizing activity against SARS-CoV-2 from a synthetic nanobody library in a timeframe of only 2-3 weeks. The traditional generation of nanobodies requires at least 6 weeks for the Llama immunization and a total of 3-4 months for the entire selection approach [33]. The sybody platform does not require any immunization steps, has a large and diverse binder repertory in order to account for the antibody maturation that naturally occurs *in vivo* and only requires a fraction of the purified antigen for the entire selection procedure [27]. The use of nanobodies as therapeutic agents has been validated by the first nanobody drug approved in 2019 [34]. Synthetic libraries could boost this new technology in the establishment of platforms for the development of nanobody-based drugs, with special importance in times that fast methodologies are necessary.”

2) how do binding sites compare to the upcoming report of the Bjorkman group mapping several human antibodies to the spike? [https://www.cell.com/cell/pdf/S0092-8674\(20\)30757-1.pdf](https://www.cell.com/cell/pdf/S0092-8674(20)30757-1.pdf)

It appears that the binding site of Sb23 to RBD is exactly opposite to the mAB_C105 binding to RBD. C105 antibody binds in the same region as the human receptor ACE2 and, as seen for ACE2, the C105 antibody can only bind to the RBD in the up conformation.

We have now added this this comparison between the Sb23 and the AB_C105 in the discussion section.

You now can read from line 236:

“A cryo-EM complex, that became available while this work was under review, between a traditional monoclonal FAB (FAB_C105) and SARS-CoV2-Spike shows binding to both a ‘2-up’ and a ‘3-up’

conformation [38]. Interestingly, FAB_C105 and Sb23 bind at two different but partially overlapping regions on the RBD (Supplementary Fig. 6). Sb23 binds on the outer RBD surface that is opposite to the FAB_C105 RBD-binding site. Thereby, Sb23 can bind the RBD in both 'up' and 'down' conformation. On the other hand, FAB_C105 can only bind to RBD in the 'up' conformation since its binding site is not available in the RBD down conformation as FAB_C105 is sterically blocked by a neighbouring FAB_C105 variable heavy chain binding an RBD in the up conformation (Supplementary Fig. 6). Arguably, this shows the strength of small high-affinity and strongly neutralising SARS-CoV-2 Spike binders such as Sb23 in that they are not as often sterically occluded in the same fashion as traditional much larger binders such as FABs."

2) What would be a next step of relevance to make use of the study?

We followed the reviewer suggestion and have now included new data to answer this question. We have identified 2 groups of binders with distinct epitopes and tested their capacity for competing with ACE2 (Supplementary Fig. 7).

From line 247, you can read: "Furthermore, among the sybodies that displayed IC₅₀ values below 20 µg/ml, we have identified a group of sybodies (Sb12, 76 and 100) that can bind simultaneously with Sb23 to RBD and were also able to abolish or decrease the binding affinity of ACE2 (Supplementary Fig. 7). For the development of therapeutic agents, it would be advantageous to further increase the avidity and affinity of these binders. Heterobivalent ligands are well known in the pharmacology area, for their distinctly increased avidity and residence time over the individual ligands [39]. By combining the neutralization potential of Sb23 with a non-overlapping sybody, we could potentially increase the overall avidity towards the spike protein. Future work could involve the creation of fusion sybodies and test their neutralization potential.

3) what is the important result of the SAXS data? It is described at length, but it is not obvious what it adds to the study.

SAXS was used for rapid quality control and characterization of the sybodies, RBD, Spike and ACE2 in solution and helped in the selection process of Sb23. The subsequent SAXS analysis showed the binding site of Sb23 close to that of ACE2 on the RBD. The SAXS studies were directly and rapidly generated without special handling or freezing and encouraged us to pursue the cryo-EM structure studies of Spike with Sb23.

3) Up-down conformations of the spike trimer are discussed quite at length but not illustrated well.

We thank the reviewer for this point and have now included a new figure 6 where we show a top view of the SARS-CoV-2 spike bound by Sb23 in both '1-up' and the '2-up' conformation. Furthermore, in the same new Figure 6 we show the Sb23 mediated blockage of ACE2 binding in the two different conformations.

minor point:

4) the sequence information was missing in the supplementary material - many blank pages are provided.

We apologize for the inconvenience. Because of its size, the table was provided as an excel file as requested by the journal. We have now converted the table to a PDF file to avoid any further problems.

Reviewers' Comments:

Reviewer #1:

Remarks to the Author:

Amendments have been incorporated to meet the comments of the reviewers.

Reviewer #2:

Remarks to the Author:

The authors have responded well to the comments - I have no further comments